# OpenReview forum: "Aligning Large Language Models through Synthetic Feedback"
_EMNLP/2023/Conference — EMNLP 2023 Main_

### Official Review · Reviewer_Zsnr · 2023-08-05

**Soundness:** 4

**Excitement:**

4: Strong: This paper deepens the understanding of some phenomenon or lowers the barriers to an existing research direction.

**Paper Topic And Main Contributions:**

This paper proposes a new method for improving the alignment of large language models, which is the process of ensuring that the model's outputs are consistent with human expectations. It presents a novel alignment learning framework that uses a synthetic preference ranking dataset to train a reward model, and leverages the feedback from that reward model to do both (reward model guided) supervised fine-tuning and reinforment learning. The resulted model, ALMoST, outperforms several recent LLMs using external human or pre-aligned LLM supervisions (Alpaca, Dolly v2, OpenAssistant) on alignment benchmarks (Static HHH, TruthfulQA) and human evaluations.

The key contribution is a simple yet intuitive way to construct the synthetic preference ranking dataset, without relying on external supervions such as human feedback or responses from pre-aligned models. Based on the assumption that larger and better prompted models are likely to produce responses with higher quality for the same query, ranking labels can be automatically obtained from synthetic responses from different models with controlled sizes (e.g. larger > smaller), in-context learning examples (more > less), and prompt quality (detailed > concise). Experimental results show that (1) the LLM trained using the feedback from the proposed reward model outperforms competitive baselines using external human or pre-aligned LLM supervisions; (2) the reward model trained on the synthetic dataset is comparable with the one trained on human preference data (HH-RLHF).

Another contribution includes the exploration of using the reward model to generate supervised fine-tuning data of better quality. A new method is proposed, namely reward-model guided self play (RMSP), where the best based model is used to generate synthetic instruction following data and the reward model is used to filter out low-preference responses. Experimental results show that the model trained using the filtered data outperforms its counterpart without using the reward model.



**Questions For The Authors:**

Question A: Relevant to the first item above, it is unclear to me why `LLaMA-30B-Faithful-3shot` is better than `LLaMA-30B-HHH-5shot`.

Question B: see the third item above

**Reasons To Accept:**

- Aligning LLMs with Human values is one of the most important research directions in the field of LLM research. Different from previous work either distills from pre-aligned LLM outputs or learns from additionally collect human feedback data, the method proposed by this paper does not require any of these external supervisions, which significantly reduce the cost of alignment learning. I believe the idea would inspire more research along the same line in the future.
- The full recipe of how to use synthetic feedback to train a better aligned model is beneficial to the community.

**Reasons To Reject:**

- The core assumption that the response from a larger LLM with more and better demonstrations might be better overall is not conprehensively studied. Although the performance of the trained reward model can be viewed as a indirect evidence supporting the assumption, it would still be benefitial to understand when and how the assumption may not hold through a systematic evaluation on the output of various LLM configurations used.
- From Table 3, the heuristic filter seems to play an important role in the synthetis data generation process. While sub-optimal, the filter requires a fair amount of expert knowledge and empirical experience to design and improve, which conflict with the motivation of reducing human effort.
- It is unclear how to use the apporach to align larger models. For example, as the supervised fine-tuning data used to train the 7B model in the paper is produced by a well-prompted 30B model, what would be the teacher model to produce data for aligning a 30B model?

**Reproducibility:**

4: Could mostly reproduce the results, but there may be some variation because of sample variance or minor variations in their interpretation of the protocol or method.

**Reviewer Confidence:**

4: Quite sure. I tried to check the important points carefully. It's unlikely, though conceivable, that I missed something that should affect my ratings.

---

> ### Author Rebuttal · Authors · 2023-08-28
>
> **[R3#1]**
>
> > The core assumption that the response from a larger LLM with more and better demonstrations might be better overall is not conprehensively studied. Although the performance of the trained reward model can be viewed as a indirect evidence supporting the assumption, it would still be benefitial to understand when and how the assumption may not hold through a systematic evaluation on the output of various LLM configurations used.
>
> > Question A: Relevant to the first item above, it is unclear to me why LLaMA-30B-Faithful-3shot is better than LLaMA-30B-HHH-5shot.
>
> Following your valuable suggestion, we attach new intrinsic evaluations to study our core assumption (rule-of-thumb). Specifically, we conduct a GPT-4 evaluation comparing our prompted generators with Alpaca-7B as we did in our paper. The table below shows that our assumption is valid in all the three factors we claimed (model size, the number of demonstrations, and the quality of demonstrations).
>
> | vs. Alpaca-7B | Win | Tie | Loss | Relative Score |
> | --- | --- | --- | --- | --- |
> | LLaMA-7B-HHH-1shot | 12 | 15 | 133 | 54.1 |
> | LLaMA-7B-HHH-3shot | 12 | 14 | 134 | 59.9 |
> | LLaMA-7B-HHH-5shot | 15 | 17 | 128 | 60.0 |
> | LLaMA-13B-HHH-3shot | 17 | 17 | 126 | 64.8 |
> | LLaMA-30B-HHH-5shot | 39 | 21 | 100 | 80.3 |
> | LLaMA-30B-Faithful-3shot | 77 | 13 | 70 | 101.0 |
> - We additionally report the Relative Score, which is a relative overall score (sum of each 1-10 score for all questions) with Alpaca-7B.
>
> Moreover, we can observe that the model size and the quality of demonstrations significantly contribute to the final response quality (LLaMA-13B-HHH-3shot << LLaMA-30B-HHH-5shot & LLaMA-30B-HHH-5shot << LLaMA-30B-Faithful-3shot), while the number of demonstrations makes a relatively small contribution, and its effect is saturated between 3 and 5 shots. Please see also R2#1 for more details. We will include this study in our final version.
>
> **[R3#2]**
>
> > From Table 3, the heuristic filter seems to play an important role in the synthetis data generation process. While sub-optimal, the filter requires a fair amount of expert knowledge and empirical experience to design and improve, which conflict with the motivation of reducing human effort.
>
> We agree that the heuristic filter is an essential pipeline in our method. However, we did not do heavy engineering for the heuristic filter to improve the performance, as shown in Figure 6. For example, we only used a few keyword-based rules as a minimal effort of qualification (If the generated response does not end with '.', If the generated response starts with 'well', If the generated response contains "I don’t know", 'image', or special-purposed prefix.). Also, we believe another assessment checking the confidence interval of response lengths is a very intuitive process that does not require expert knowledge.
>
> **[R3#3]**
>
> > It is unclear how to use the apporach to align larger models. For example, as the supervised fine-tuning data used to train the 7B model in the paper is produced by a well-prompted 30B model, what would be the teacher model to produce data for aligning a 30B model?
>
> We believe the 30B model could also be aligned with our synthetic dataset generated by the 30B model (even smaller models). In Table 4 (RMSP vs. Self-play), we found that the ability to discriminate between better or worse responses contributes to well-aligned performance. Moreover, as we see in R3#1, the quality of demonstrations (Faithful >> HHH) is a more important factor than the model size in producing high-quality responses. In other words, we can effectively contrast the good and bad responses with various combinations of demonstrations to obtain synthetic feedback.
>
> In addition to our speculation, we additionally conduct a new experiment to train a larger SFT model with the same synthetic dataset used in our paper. Specifically, we train LLaMA-30B with LoRA-tuning [1] on the same demonstration dataset used in our paper. We use the same configurations of 7B training except for the global batch size, which is set to 64 on 8 * A100 GPUs. Also, we set additional configurations (lora_r=8, lora_alpha=16, lora_dropout=0.1, target_modules=['k', 'v']) for the LoRA-tuning. We report a GPT-4 evaluation result against Alpaca-7B.
>
> | vs. Alpaca-7B | Win | Tie | Loss | Relative Score |
> | --- | --- | --- | --- | --- |
> | ALMoST-7B-SFT (Full Fine-tuning) | 87 | 17 | 56 | 104.5 |
> | ALMoST-30B-SFT (LoRA) | 109 | 20 | 31 | 112.8 |
>
> As shown in the above table, our synthetic demonstrations dataset could be used to align the 30B SFT model successfully. While this result might be sub-optimal since it is based on LoRA-tuning instead of full fine-tuning, we think fine-grained prompt design per task type instead of the static 3-shot of demonstrations will make the synthetic dataset more powerful. We remain the investigation for future work.
>
> [1] LoRA: Low-Rank Adaptation of Large Language Models (Hu et al., 2021)

---

### Official Review · Reviewer_Pu1N · 2023-08-09

**Soundness:** 4

**Excitement:**

4: Strong: This paper deepens the understanding of some phenomenon or lowers the barriers to an existing research direction.

**Paper Topic And Main Contributions:**

This paper proposes a new method to align large language models with human values using synthetic feedback. The authors follow a rule of thumb that the content generated by larger models with richer demonstrations would have better quality. Based on it, they design a pipeline to generate synthetic feedback with models of different sizes given different demonstration numbers, and then adopt the ranked preference modeling objective to train a reward model. Additionally, they propose a post validation method to guarantee the quality of the generated feedback in the reward modeling process. In the supervised fine-tuning stage, they adopt RM-guided Self-Play (RMSP), which includes the trained reward model trained in the data generation loop. In the experiment section, they demonstrate the effectiveness of their proposed method in aligning the model with human preference. Their models outperform the models trained by aligned language models or human feedback. They also conduct a series of ablation studies to validate the post validation and RMSP they designed.

**Questions For The Authors:**

1. In the reward modeling process, you train a LLaMA-7B as the reward model with LLaMA-7B, LLaMA-13B and LLaMA-30B as feedback generators. Does it imply your method requires larger feedback generators than the reward model? Have you tried to train a reward model with feedback generators comparable to or smaller than it? Like LLaMA-30B as the reward model and LLaMA-7B/-13B/-30B as feedback generators.

**Reasons To Accept:**

1. Their method is novel and interesting. The rule of thumb they follow to do reward modeling sounds reasonable. Also, other enhancements they implement, such as post validation and RMSP integration in other segments of the pipeline, prove to be quite useful.
2. The experiment results seem promising. It shows the models trained by their methods can surpass even larger models trained through genuine human feedback or other aligned models.

**Reasons To Reject:**

1. The separate ablation study of the assumptions they followed is missing. The rule of thumb they followed contains two parts: (1) feedbacks generated by larger models have better quality. (2) feedbacks generated given more demonstrations have better quality. While they do empirically verify the efficacy of jointly incorporating these two assumptions to generate synthetic feedback and train the reward model, the specific impact of each assumption on the reward model in isolation remains unclear. For example, how the reward model would behave when only trained on feedback generated given different numbers of demonstrations, but all by the same model?
2. The SFT and RLSF steps seem a little trivial as the reward model already achieves quite high scores in many aspects of human values according to Table 1.

**Reproducibility:**

3: Could reproduce the results with some difficulty. The settings of parameters are underspecified or subjectively determined; the training/evaluation data are not widely available.

**Reviewer Confidence:**

4: Quite sure. I tried to check the important points carefully. It's unlikely, though conceivable, that I missed something that should affect my ratings.

---

> ### Author Rebuttal · Authors · 2023-08-28
>
> **[R2#1]**
>
> >  1. The separate ablation study of the assumptions they followed is missing. The rule of thumb they followed contains two parts: (1) feedbacks generated by larger models have better quality. (2) feedbacks generated given more demonstrations have better quality. While they do empirically verify the efficacy of jointly incorporating these two assumptions to generate synthetic feedback and train the reward model, the specific impact of each assumption on the reward model in isolation remains unclear. For example, how the reward model would behave when only trained on feedback generated given different numbers of demonstrations, but all by the same model?
>
> As you suggested, we report additional intrinsic evaluations to investigate the impact of each assumption. To this end, we conduct a GPT-4 evaluation (on Vicuna Questions) to identify response quality from our prompted response generators compared to Alpaca-7B, as we originally did in our paper. We report the Win/Tie/Loss and Relative Score, the relative overall score (sum of each 1-10 score for all questions) over the questions compared to Alpaca-7B. We isolate the three factors,(1) model size, (2) the number of demonstrations, and (3) the quality of demonstrations, to analyze our rule-of-thumb comprehensively.
>
> First, we can see that the model size significantly contributes to the quality of response generation. The winning rate against Alpaca-7B increases monotonically as we increase the model size. The gap between 13B and 30B is especially large.
>
> | vs. Alpaca-7B | Win | Tie | Loss | Relative Score |
> | --- | --- | --- | --- | --- |
> | LLaMA-7B-Faithful-3shot | 52 | 14 | 94 | 82.7 |
> | LLaMA-13B-Faithful-3shot | 59 | 19 | 82 | 88.9 |
> | LLaMA-30B-Faithful-3shot | 77 | 13 | 70 | 101.0 |
>
> Second, we find that the number of demonstrations also contributes to the response quality, but improvements by this factor are relatively small. The performance gain is saturated between 3 shots and 5 shots when we compare them with the relative score.
>
> | vs. Alpaca-7B | Win | Tie | Loss | Relative Score |
> | --- | --- | --- | --- | --- |
> | LLaMA-7B-HHH-1shot | 12 | 15 | 133 | 54.1 |
> | LLaMA-7B-HHH-3shot | 12 | 14 | 134 | 59.9 |
> | LLaMA-7B-HHH-5shot | 15 | 17 | 128 | 60.0 |
>
> Finally, we confirm the quality of demonstrations is the most important factor, as we reported in Table 4 of our paper. Surprisingly, we find that the smaller model with the well-designed prompt (LLaMA-7B-Faithful-3shot) outperforms the larger model with the normal prompt (LLaMA-30B-HHH-5shot).
>
> | vs. Alpaca-7B | Win | Tie | Loss | Relative Score |
> | --- | --- | --- | --- | --- |
> | LLaMA-30B-HHH-5shot | 39 | 21 | 100 | 80.3 |
> | LLaMA-30B-Faithful-3shot | 77 | 13 | 70 | 101.0 |
>
> We appreciate your constructive suggestion and will surely include these analyses in the final version.
>
> **[R2#2]**
>
> > 2. The SFT and RLSF steps seem a little trivial as the reward model already achieves quite high scores in many aspects of human values according to Table 1.
>
> Yes, we agree with you. Our main contribution stems mostly from the reward modeling stage (Step 1) since the remaining stages are not feasible without synthetic feedback from the first stage. Even without the remaining two stages, we could perform adequate inference with the reward model, e.g., best-of-N sampling with prompted baseline model (LLaMA-30B-Faithful-3shot). However, we think the remaining stages are also important to build an effective and efficient assistant model. First, the SFT stage makes the policy model standalone by eliminating dependency on the demonstrations. In other words, the model’s efficiency could be enhanced because of the reduced token consumption for the demonstrations in long sequences. Second, the RLSF stage also discards the dependency on the reward model at inference time. As a result, we conclude the SFT and RLSF stages are still needed to effectively transfer the reward model's ability into the policy model.
>
> **[R2#3]**
>
> > 1. In the reward modeling process, you train a LLaMA-7B as the reward model with LLaMA-7B, LLaMA-13B and LLaMA-30B as feedback generators. Does it imply your method requires larger feedback generators than the reward model? Have you tried to train a reward model with feedback generators comparable to or smaller than it? Like LLaMA-30B as the reward model and LLaMA-7B/-13B/-30B as feedback generators.
>
> No, we do not think leveraging larger generators is an essential requirement for our reward modeling. As we showed in R2#1, the quality of demonstrations contributes to the response quality more than the model size. On the other hand, we think the various-sized large models could be used to improve overall synthetic feedback quality. We have not tried the 30B RM reward model training yet because of resource constraints, but we would like to investigate the generalizability according to model size in future work.

---

### Official Review · Reviewer_NN7k · 2023-08-10

**Typos Grammar Style And Presentation Improvements:** 1. On the first look, the term “pre-a…
**Soundness:** 3

**Excitement:**

3: Ambivalent: It has merits (e.g., it reports state-of-the-art results, the idea is nice), but there are key weaknesses (e.g., it describes incremental work), and it can significantly benefit from another round of revision. However, I won't object to accepting it if my co-reviewers champion it.

**Paper Topic And Main Contributions:**

This paper proposes a novel Reinforcement Learning from Synthetic Feedback (RLSF) framework to align a vallina LM, LLaMA-7B, without any LLM API distillations or intensive human annotated demonstrations leveraging responses generated by LMs of different sizes/shots under the assumption that response from larger LMs with more shots might
be better overall.

The results suggest their resulting aligned LM — Aligned Language Model with Synthetic Training dataset (ALMoST) — achieves superior performances compared to other open-sourced aligned LMs in both alignment benchmarks and human evaluations.

**Reasons To Accept:**

1. Address the challenge of requiring LLM API or human annotated demonstrations to align Valinna LM.

2. Thorough evaluation/ablation study of the RM.

3. The paper is generally well-written with clear structures.

**Reasons To Reject:**

1. The key role-of-thumb hypothesis — responses from larger LLMs with more demonstrations might be better overall — alone does not deliver ideal synthetic feedback for effective RM learning. Without the Heuristic Filter post validation, the proposed RM even falls short of the simple lengthy baseline (Table 3).

2. The training of As-is RM requires 20k stackexchange ranking pairs. This seems to contradict the main goal of the paper, i.e., minimizing reliance on human annotated data.

3. Incorporating the ALMoST (RM) in the results of alignment benchmark in Table 1 is not suitable, as the goal is to evaluate different aligned LMs, instead of the reward model. When excluding the ALMoST (RM), the performance of RLSF aligned LM, i.e., the proposed ALMoST (PPO), is notably worse than Vicuna which doesn’t require comparison data to train.


**Reproducibility:**

3: Could reproduce the results with some difficulty. The settings of parameters are underspecified or subjectively determined; the training/evaluation data are not widely available.

**Reviewer Confidence:**

3: Pretty sure, but there's a chance I missed something. Although I have a good feel for this area in general, I did not carefully check the paper's details, e.g., the math, experimental design, or novelty.

---

> ### Author Rebuttal · Authors · 2023-08-28
>
> **[R1#1]**
>
> > 1. The key role-of-thumb hypothesis — responses from larger LLMs with more demonstrations might be better overall — alone does not deliver ideal synthetic feedback for effective RM learning. Without the Heuristic Filter post validation, the proposed RM even falls short of the simple lengthy baseline (Table 3).
>
> We include the new intrinsic evaluation to probe our rule-of-thumb hypothesis. For this, we conduct a GPT-4 evaluation comparing the prompted generators in our configurations against Alpaca-7B, as we did in our paper. Note that we do not perform any post validation for this. We can observe that our rule-of-thumb hypothesis is still valid apart from using the heuristic filter in this evaluation. As shown in the table below, the overall generation qualities based on winning rate and relative score are matched with our rule-of-thumb.
>
> | vs. Alpaca-7B | Win | Tie | Loss | Relative Score |
> | --- | --- | --- | --- | --- |
> | LLaMA-7B-HHH-1shot | 12 | 15 | 133 | 54.1 |
> | LLaMA-7B-HHH-3shot | 12 | 14 | 134 | 59.9 |
> | LLaMA-13B-HHH-3shot | 17 | 17 | 126 | 64.8 |
> | LLaMA-30B-HHH-5shot | 39 | 21 | 100 | 80.3 |
> | LLaMA-30B-Faithful-3shot | 77 | 13 | 70 | 101.0 |
> - We additionally report the Relative Score, which is a relative overall score (sum of each 1-10 score for all questions) with Alpaca-7B.
>
> Nevertheless, as you pointed out, we found the RM trained without the post validation significantly failed to choose a better response. We presume the reward modeling would be a sensitive process vulnerable to noisy signals. As we explained in L164, our assumptions are not always valid because of the stochastic nature of LLMs (as we conduct top-p=0.9 sampling for the response generation). Thus, we think that our proposed filtration method could support our hypothesis effectively removing such noises in our framework, making the reward modeling more robust as a result. We will include this new evaluation and add more explanation for the necessity of the post validation in the final version.
>
> **[R1#2]**
>
> > 2. The training of As-is RM requires 20k stackexchange ranking pairs. This seems to contradict the main goal of the paper, i.e., minimizing reliance on human annotated data.
>
> The distinction between pretraining data (or web crawl data) and human-annotated data could be often unclear. We consider Stack Exchange to be pretraining data given that it is crawled from the raw web since comparison datasets (question, chosen answer, rejected answer) could be automatically constructed using the meta information naturally obtained by the real users, e.g., upvote or label for the accepted answer.  For instance, [1] also uses the Stack Exchange dataset as a pretraining dataset, along with Reddit comments and Wikipedia edits, for their reward modeling. Also, note that our reward model still performs competitively even without the As-is RM filtration, as shown in Table 3.
>
> [1] A General Language Assistant as a Laboratory for Alignment (Askell et al., 2021)
>
> **[R1#3]**
>
> > 3. Incorporating the ALMoST (RM) in the results of alignment benchmark in Table 1 is not suitable, as the goal is to evaluate different aligned LMs, instead of the reward model. When excluding the ALMoST (RM), the performance of RLSF aligned LM, i.e., the proposed ALMoST (PPO), is notably worse than Vicuna which doesn’t require comparison data to train.
>
> Thank you for your suggestion. We intended to show the performance of our RM along with other generator models to emphasize that the adequate performances of our SFT and PPO models are derived from the well-performed RM. However, we agree that we could have made our presentation better to prevent misunderstanding and we will make Table 1 more clear as you suggested.
>
> **[R1#4]**
>
> > 1. On the first look, the term “pre-aligned” LLMs is confusing as readers may refer to it as vanilla LMs. Also, there are some inconsistencies in the usage of the term, e.g., in Line 391 and 425 it becomes “aligned” LLMs.
>
> > 2. Moving Line 125 ~ 127 into the introduction section could be beneficial for readers to understand the proposed framework more quickly with less confusion.
>
> Thank you for your valuable suggestions. We will change the term "pre-aligned" LLMs to an unambiguous term. Also, we will reflect on your other suggestions in our final version.

---

### Meta-Review · Area_Chair_nitN · 2023-09-20

**Recommendation:** 4

**Metareview:**

The paper presents a novel alignment learning framework that relies on synthetic feedback rather than extensive human annotations or pre-aligned large language models. By utilizing reward modeling with synthetic feedback and reinforcement learning, the proposed method shows improved performance in alignment benchmarks compared to recent open-sourced models. The reviewers are generally positive towards the contribution of this paper. I therefore would recommend acceptance of this paper to the main conference.

---

### Decision · Program_Chairs · 2023-10-07

**Decision:**

Accept-Main

**Comment:**

The paper presents a novel alignment learning framework that relies on synthetic feedback rather than extensive human annotations or pre-aligned large language models. By utilizing reward modeling with synthetic feedback and reinforcement learning, the proposed method shows improved performance in alignment benchmarks compared to recent open-sourced models. The reviewers are generally positive towards the contribution of this paper. I therefore would recommend acceptance of this paper to the main conference.